# Push Past Green: Learning to Look Behind Plant Foliage by Moving It

**Xiaoyu Zhang**
University of Illinois at Urbana-Champaign
zhang401@illinois.edu

**Saurabh Gupta**
University of Illinois at Urbana-Champaign
saurabhg@illinois.edu

**Abstract:** Autonomous agriculture applications (*e.g.*, inspection, phenotyping, plucking fruits) require manipulating the plant foliage to look behind the leaves and the branches. Partial visibility, extreme clutter, thin structures, and unknown geometry and dynamics for plants make such manipulation challenging. We tackle these challenges through data-driven methods. We use self-supervision to train SRPNet, a neural network that predicts what space is revealed on execution of a candidate action on a given plant. We use SRPNet with the cross-entropy method to predict actions that are effective at revealing space beneath plant foliage. Furthermore, as SRPNet does not just predict how much space is revealed but also where it is revealed, we can execute a sequence of actions that incrementally reveal more and more space beneath the plant foliage. We experiment with a synthetic (vines) and a real plant (Dracaena) on a physical test-bed across 5 settings including 2 settings that test generalization to novel plant configurations. Our experiments reveal the effectiveness of our overall method, PPG, over a competitive hand-crafted exploration method, and the effectiveness of SRPNet over a hand-crafted dynamics model and relevant ablations. Project website with execution videos, code, data, and models: https://sites.google.com/view/pushpastgreen/.

**Keywords:** Deformable Object Manipulation, Model-building, Self-supervision

## 1 Introduction

The ability to autonomously manipulate plants is crucial in the pursuit of sustainable agricultural practices [1, 2, 3, 4]. Central to autonomous plant manipulation is the *plant self-occlusion problem*. Plants self-occlude themselves (Figure 1 (left)). Plant leaves and branches have to be carefully moved aside for the simplest of agriculture problems: plant inspection, phenotyping, precision herbicide application, or finding and plucking fruits. This papers tackles this plant self-occlusion problem. We develop methods that learn to manipulate plants so as to look beneath their external foliage. Figure 1 (middle and right) shows steps from a sample execution from our method. We believe our work will serve as a building block that enables many different applications that require manipulation of plants in unstructured settings.

Manipulating external plant foliage to reveal occluded space is hard. Sensing is difficult because of dense foliage, thin structures and partial observability. Control and planning is challenging because of unknown dynamics of the plant leaves and branches, and the difficulty of building a full articulable plant model. These sensing and control challenges motivate the need for learning. However, use of typical learning paradigms is also not straight-forward. Model-free RL (*e.g.* PPO [5]) requires interaction data at a scale that is difficult to collect in the real world. Model-based RL is more sample-efficient, but is quite challenging here as precisely predicting the next observation (or state) is hard. Imitation learning is more promising; but for the exploration task we tackle, the next best action depends on what has already been explored. This increases the amount of demonstration data required to train models. Lack of high-fidelity plant simulators preclude simulated training.

Our proposal is to tackle this problem through self-supervision [6, 7]. We collect a dataset of action outcomes (amount of space revealed) by letting the robot randomly interact with plants. We use this data to train a model to predict space revealed by an input action. However, in order to derive a long-term strategy for exploring all of the space beneath the plant, the model has to predict not

7th Conference on Robot Learning (CoRL 2023), Atlanta, USA.

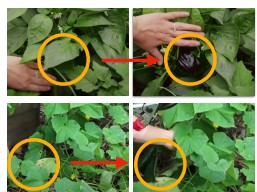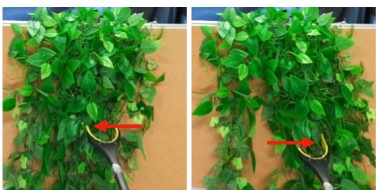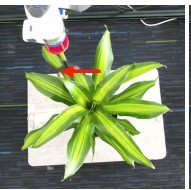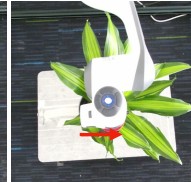

**Figure 1: (left)** Plants self-occlude themselves. Two examples of leaves and branches being pushed aside for inspection and picking fruits. This paper develops learning algorithms that enable robots to tackle this plant self-occlusion problem. We show actions executed by the robot to expose the space behind vines **(middle)** and Dracaena plant **(right)**.

only how much space would get revealed, but also *where* (Figure 2 (b), Section 4.1). In this way, the model output lets us reason about what *additional* space each action would reveal. This allowing us to execute multi-step action sequences that explore all of the area behind the plant using a simple greedy control loop implemented via the cross-entropy method (CEM) (Figure 2 (a), Section 4.3).

This paper implements and tests these ideas on a physical platform that is tasked with revealing space behind decorative vines and a real Dracaena plant. We collect 48 hours of plant interaction data and use it to train a neural network that we call *Space-Revealed Prediction Network (SRPNet)*. SRPNet, when used with CEM, leads to effective control strategies to reveal all (or user-specified) space beneath the plant foliage. We call our overall framework *PushPastGreen* (PPG).

Experiments show that SRPNet outperforms a hand-crafted dynamics model and ablated versions of SRPNet. In physical experiments, PPG outperforms a hand-crafted exploration strategy and versions of PPG that replace SRPNet with alternative choices for modeling space revealed. In all 5 settings across vines and Dracaena, including 2 that explicitly test for generalization, we observe relative improvements ranging from 4% to 34% over the next best method. This establishes the benefits of PPG and the use of learning to manipulate plants.

## 2 Related Work

**Autonomous Agriculture.** Motivated by the need for adopting sustainable agricultural practices [2, 3, 1], researchers have sought to introduce and expand the use of autonomy for agricultural tasks [8, 9]. While a full review is beyond our scope, major trends include a) development of specialized robotic hardware [10, 11, 12], b) development of algorithms for perception in cluttered agricultural settings [13, 14, 15], c) design of control algorithms for navigation [16, 17] and manipulation [18], and d) full autonomous farming systems [19, 18, 20, 21].

**Plant Manipulation.** For manipulation oriented tasks (*e.g.* fruit picking): [22] compute 3D grasp pose for largely unoccluded fruits, [23] design a visual servoing approach to get partially occluded fruits into full view, [24, 25, 26] output trajectories for reaching fruits while avoiding collisions with plant leaves and branches, and [10, 27] develop soft arms / end-effectors that can maneuver around plant structures. Much less research actually interacts with the plant structure to accomplish tasks. [18] hand-design strategies for pushing fruits out of the way. [28] show simulated results using probabilistic motion primitives for pushing fruits out of the way. We instead study the task of looking behind plant foliage, and hand-crafted strategies proposed in [28, 18] are not directly applicable to our setting. [29, 30] tackle reaching in plants while treating leaves as permeable obstacles, while [31] develops efficient MPC to minimize contact forces when interacting with plants. [32] learn to model object's resistance to movement by estimating stiffness distribution from torque measurements. We instead directly model the effect of actions executed on the plant.

**Manipulation of Deformable Objects.** Past works have considered manipulation of other deformable objects such as cloth [33, 34, 35, 36, 37], ropes [38, 39], elasto-plastics [40, 41], fluids [37, 42, 43], and granular media [44]. [33, 34, 38] design dynamic primitive actions to tackle cloth and rope manipulation. [45, 42, 40, 35] learn particle-based forward models for deformable material and use model-based RL for control. [46, 41] compose skills for deformable object manipulation to solve long-horizon tasks. Our study explores plant manipulation. Lack of high-fidelity plant simulators limits the applicability of past methods that rely on large amount of data in simulation [42, 36, 38]. At the same time, building dynamics models [40, 35] for plants is hard due to dense foliage, thin branch structure, and unknown heterogeneous dynamics.

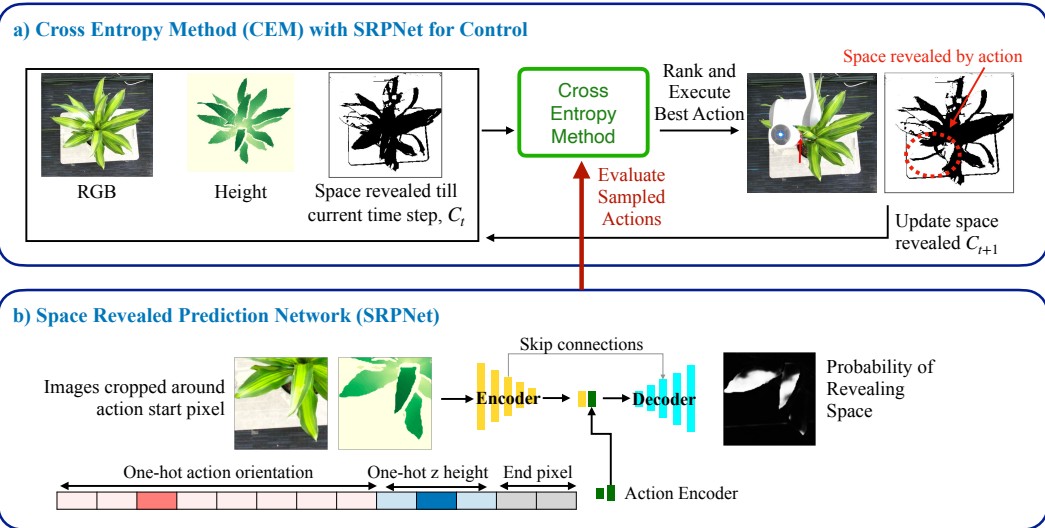

**Figure 2: Overview of PushPastGreen.** PushPastGreen learns to manipulate plants to reveal the space behind them thus tackling the plant self-occlusion problem. PushPastGreen includes Space-Revealed Prediction Network (SRPNet) that predicts where space is revealed upon execution of a pushing action, as shown in **(b)** and described in Sec. 4.1. SRPNet can not only rank actions based on how much space they will reveal, but because it can also predict *where* space gets revealed, it can also be used for executing multi-step trajectories that explore all the space behind the vines as shown in **(a)** and described in Sec. 4.3. SRPNet is trained using self-supervision as described in Sec. 4.2.

**Self-supervised Learning in Robotics.** We adopt a self-supervised approach for training our models. Self-supervision techniques typically predict scalar quantities (*e.g.* grasp outcomes [6, 7], delta cloth coverage on workspace [33], pushing+grasping success [47], *etc.*). Past work has also used self-supervision to build forward models for model-predictive control [48, 49, 50, 51] in pixel or feature spaces. Our work finds a middle ground. We predict not just how much space is revealed (insufficient for executing a sequence of actions), but also where it is revealed. This lets us execute sequences of actions that incrementally expose more and more space.

## 3 Problem Setup

Figure 3 shows the 2 different plants that we tackle, a) decorative vines vertically hanging across a board, and b) a real Dracaena plant. The vines involve a 2D exploration problem and present challenges due to entanglement, thin structures, and extensive clutter. The real Dracaena plant exhibits a large variation in scene depth leading to a 3D problem. The Dracaena plant has big leaves that bend only in specific ways. Thus it requires careful action selection. Both test cases exhibit unknown and heterogeneous dynamics which makes it hard to manipulate them.

As one can notice in Figures 1 and 3, vines occlude the surface behind the vines. Similarly, the Dracaena leaves occlude the plant. We refer to this occlusion as the *plant self-occlusion problem*. The task is to have manipulation policies that can use the non-prehensile pushing actions (as described below) to reveal the space beneath the plant surface.

We use the Franka Emika robot and change the end effector to a grabber (as also done in past work [52, 53]). We use RGB-D cameras pointed at the plant for sensing. Our action space consists of non-prehensile planar pushing actions (also used in past work *e.g.* [51]). We sample a 3D location and push in a plane parallel to the board for the vines and to the ground for the Dracaena plant. As vines have limited depth variation, we use a fixed $z$ for the vines, but actions are sampled at varying $z$ for the Dracaena plant. Sections A.1 and B.1 provide more experimental details.

## 4 Proposed Approach: Push Past Green

PPG adopts a greedy approach. We keep track of space that has not yet been revealed, and execute actions that would reveal the most *new* space. Doing this requires a model that predicts what space a

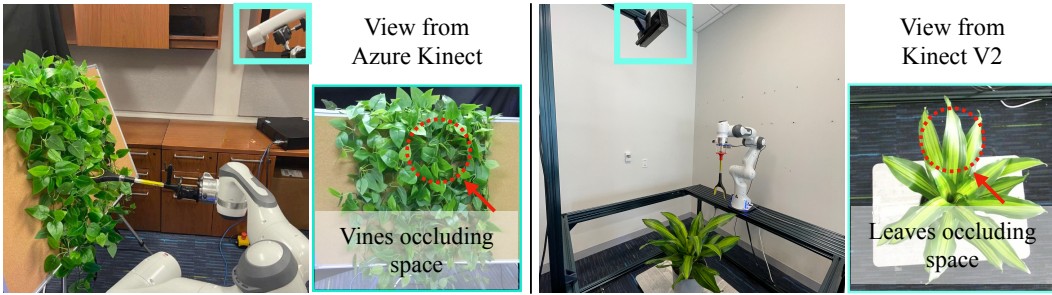

**Figure 3: Hardware setup for vines (left) and real Dracaena plant (right).** We use a grabber as the end-effector [52, 53]. View from the RGB-D camera is in the inset. The task is to move the vines and the Dracaena leaves aside to reveal the space occluded by them.

candidate action would reveal. As plants are complex to model, such a model is hard to hand-craft. Furthermore, it is difficult to estimate the precise state and physical parameters for plants from a single RGB-D image (*e.g.* placement of leaves and branches with respect to one another, location and connectivity of all the leaves with the stems, stiffness parameters). This precludes the use of physical simulation for such prediction. Thus, we design Space-Revealed Prediction Network (SRPNet), that uses learning to directly predict space revealed on execution of a given action on a given plant configuration (Section 4.1). Learning to directly make this prediction sidesteps the complexity of precise state estimation and physical simulation necessary to build a full dynamics model for the plant. To obtain the data to train SRPNet, we adopt a self-supervised approach and execute random actions from the robot action space. We automatically compute the space revealed after an action using the RGB-D image (Section 4.2). Together with SRPNet, we design PPG, a greedy algorithm that uses the cross-entropy method (CEM) [54] to sample the action that promises to reveal the most *new* space on top of space already revealed (Section 4.3).

### 4.1 Space-Revealed Prediction Network

**Input Representation.** As shown in Figure 2 (b), the input to our model is a $200 \times 200$ patch cropped out at the action start location. We crop both the RGB image and an image denoting the height relative to the surface beneath the vines (or relative to the ground beneath the Dracaena). The height image is computed using the point cloud from the RGB-D cameras. As the model sees crops around the site of interaction, each action starts at the center of the image. We only need to represent the $z$ coordinate of the push start location, the push direction ($\theta$), and the push distance ($d$). We represent these using a) a one-hot vector depicting the push direction, b) a one-hot $z$ height, and c) push distance via the location of the action end-point *i.e.* $[d\cos(\theta), d\sin(\theta)]$.

**Output.** The model produces an output that is the same size as the input. Each value in this spatial map represents the probability that space will get revealed at that location in the image upon execution of input action on the input plant configuration.

**Model Architecture and Loss.** We adopt the UNet structure [55] used in image segmentation. The encoder has 5 convolution layers. The action features are processed through 2 transposed convolution layers before being concatenated with the visual features and passed to the decoder with 5 transposed convolution layers. We add a skip connection between each corresponding convolutional and transposed convolution layer. SRPNet is trained using cross-entropy loss.

### 4.2 Data Collection and Preparation

Our self-supervised data collection procedure executes random actions from the robot action space. We divide the robot's reachable space into a grid of 2cm $\times$ 2cm cells. Action starting locations $(x, y, z)$ are sampled at the centers of these cells. We sample push directions and push by 15cm clipping to the feasible space as necessary. Each interaction executes in about 30s. We record RGB-D videos and robot end-effector pose over the entire duration of the interaction. We collected 3529 interactions for vines over 30 hours, and 2175 interactions for Dracaena over 18 hours. We split the dataset into train, val, and test splits in a 8:1:1 ratio and train one model for each plant.

We automatically compute ground truth for training the model on the collected data. This involves processing the RGB and depth image before and after the interaction. For vines, we found a simple decision rule using the color value and change in depth to work well. For Dracaena, very often the entire plant wobbles upon interaction, which leads to erroneous estimates. Thus, we first align the point clouds before and after interaction and then look for depth increase to obtain ground truth. More details are provided in Supplementary Sections A.3 and B.3.

### 4.3 Looking Behind Leaves Using SRPNet

Algorithm 1 describes our control algorithm that uses the trained SRPNet to predict actions to reveal space behind vines. At each timestep $t$ of the trajectory, we use the cross-entropy method (CEM) [54] to pick out the best action to execute (line 4). We maintain the revealed space so far ($C_t$). $C_0$ is initialized to be the space visible before any actions (line 1). Action parameters are sampled from Gaussian distributions. For each candidate action, SRPNet

---

**Algorithm 1**: PPG: Revealing space beneath plants.

**Require:** Model $f$ that predicts space revealed after action
1: Current revealed space, $C_0 \leftarrow$ space visible at start
2: **for** $t \leftarrow 0$ to $T-1$ **do**
3:      Receive images $I_t$
4:      $a_t \leftarrow \text{CEM}(C_t, f, I_t)$
5:      Execute action $a_t$
6:      Calculate additional space revealed $c_t$
7:      Update current revealed space: $C_{t+1} \leftarrow C_t \cup c_t$
8: **end for**

---

predicts where space would be revealed. We determine *new* space revealed by subtracting the area that has already been revealed ($C_t$) from SRPNet's output. Samples that are predicted to reveal the most *new space* are selected as elite which are used to fit a Gaussian distribution to sample actions for the next CEM iteration. After all iterations, CEM outputs $a_t$, the action found to reveal the most new space (line 4). Upon executing $a_t$, we observe the space that is actually revealed and update $C_t$ (line 6 and 7). The process is repeated for the length of the trial.

## 5 Experiments and Results

We test our proposed framework through a combination of offline evaluations of SRPNet on our collected dataset (Section 5.1), and online execution on our physical platform for the task of revealing space behind plants (Section 5.2). Our experiments evaluate a) the benefit of learning to predict space revealed by actions, b) the effectiveness of SRPNet's input representation, and c) the quality of SRPNet's spatial predictions and selected actions for long-horizon and targeted exploration.

### 5.1 Offline Evaluation of SRPNet

We train and evaluate SRPNet on data gathered on our physical setup as described in Section 4.2. We measure the average precision (AP) for the pixels labeled as revealed-space. We train on the train split, select model checkpoints on the validation set, and report performance on the test set in Table 1. For vines, we report performance in two settings: Vines [All] *i.e.* seeing the board (behind the vines) counts as revealed space, and Vines [5cm] *i.e.* height decrease of 5cm counts as revealed space. For Dracaena, only seeing past the leaf (as determined by our automated processing from Section 4.2) counts as revealed space.

| Methods | Vines [All] | Vines [5cm] | Dracaena |
|---|---|---|---|
| Full SRPNet (Our) | 46.3 | **54.4** | **44.2** |
| *Input Representation Ablations* | | | |
|     No action | 30.2 | 43.5 | 28.4 |
|     No height map | **46.9** | 49.1 | 40.6 |
|     No RGB | 33.4 | 46.4 | 28.7 |
|     No RGB and no height map | 28.4 | 35.2 | 10.5 |
| *Data Augmentation Ablations* | | | |
|     No left/right flips | 44.1 | 52.7 | 34.6 |
|     No color jitter | 41.0 | 51.4 | 30.7 |

**Table 1: Average precision for different models at predicting space revealed.** Higher is better. Our proposed input representation outperforms simpler alternatives and data augmentation boosts performance.

**Results.** Experiments presented in Table 1 reveal the effectiveness of our method and provide insights about the underlying data. **First**, across all three settings, the full model is able to extract information from visual observations to produce higher quality output than just basing the predictions on the action information alone (*Full model vs. No RGB and no height map*). This suggests that plant configuration (as depicted in the visual observations) is important in predicting the action

outcome. **Second**, use of action information leads to better predictions (*Full model vs. No action*). This suggests that different actions at the same site produce different outcomes, and SRPNet is able to make use of the action information to model these differences. **Third**, looking at the performance across the three settings, both height map and RGB information are useful for accurate predictions. There are no trivial solutions of the form 'space gets revealed where height is high'. **Fourth**, as we only have a limited number of training samples, data augmentation strategies are effective. Supplementary Figure S3 visualizes the predictions from different versions of our model on samples from the test set. Note the nuances that our model is able to capture in contrast to the ablated versions.

## 5.2 Online Evaluation for Looking Behind Plants Task

We next measure the effectiveness of our proposed framework (PPG w/ SRPNet) for the task of looking behind plant surface (as introduced in Section 3). We measure the space revealed in units of $cm^2$ for vines and number of pixels for Dracaena.[1]

We start out by demonstrating that PPG w/ SRPNet is able to differentiate between good and bad actions. We then demonstrate our method on the task of looking behind plants over a 10 time-step horizon. Finally, we tackle the task of revealing space behind a user-specified spatial target. We conduct experiments on both the vines and the Dracaena plant. As plants can't be exactly reset, all experiments test generalization to some extent. To further test generalization, we explicitly evaluate performance on plant configurations that differ from those encountered during training.

Inexact resets also pose a challenge when comparing methods. We randomly reset the plant (such as rotating the Dracaena) between trials and expect the variance due to inexact resets to average out over multiple trials. To prevent experimenter or other unknown environmental bias, we a) randomly interleave trials for different methods, and b) reset *before* revealing which method runs next.

### 5.2.1 Baselines

**Tiling Baseline.** For the long-horizon exploration tasks, this hand-crafted baseline randomly samples from action candidates that are spread out across the workspace as shown in Figure 4. We aid this baseline by limiting the action candidates to be horizontal pushes for the vines and tangential actions for the Dracaena. We found these actions to be more effective than actions in other orientations, see Table S1 for vines and Table S3 for Dracaena.

**PPG w/ Other Dynamics Models.** To disentangle if the improvement is coming from our learned SRPNet or simply from keeping track of space that has already been revealed ($C_t$) in PPG, we swap SRPNet for other models in PPG. Specifically, we compare to a hand-crafted dynamics model (described below) and the SRPNet No Image (*i.e.* no RGB and no height map) model from Table 1.

We construct **hand-crafted forward models** for vines and the Dracaena plant. These baseline models represent vines as vertically hanging spaghetti, and the Dracaena plant as 2D radially emanating spaghetti. Figure 5 shows the the induced free space upon action execution for this baseline model.

### 5.2.2 Results

**Single Action Selection Performance.** Table 2 compares the effectiveness of PPG w/ SRPNet at picking an action that reveals the most occluded space, against random actions from the robot's action space. For the strongest comparison, we limit the random sampling to the most effective actions, horizontal pushes for the vines and tangential pushes for the Dracaena, as shown in Figure 4. Table 2 reports the average space revealed

| Method | Area Revealed | |
| --- | --- | --- |
| | Vines ($cm^2$) | Dracaena (pixels) |
| Random Horizontal / Tangential Action | 211.7 [184.2, 255.8] | 5125.6 [3423.8, 7147.0] |
| PPG w/ SRPNet (Our) | **344.1** [320.4, 372.3] | **7644.4** [6335.6, 9057.5] |

**Table 2:** PPG selected actions are more effective at revealing space.

(along with 95% confidence interval) over at least 20 trials for each method. Our approach leads to a relative improvement of 62% for vines and 49% for Dracaena over this strong baseline. This suggests that our model is able to interpret visual observations to identify good interaction sites.

---

[1]The criterion to automatically determine revealed space occasionally fails. We manually inspected test runs to confirm and fix the output of the automated method. Note, that this manual inspection is done during evaluation only. No method has access to such manual inspection, neither during training nor during execution.

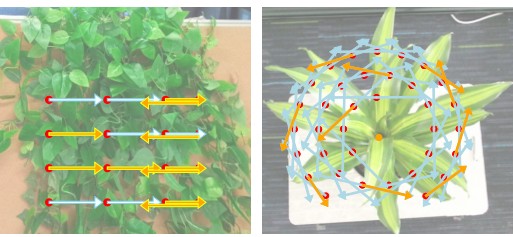
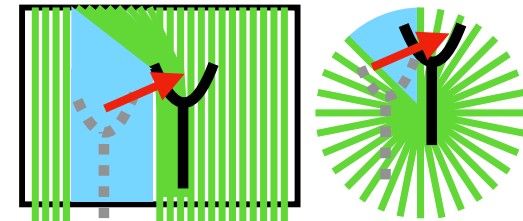

**Figure 4: Tiling baseline.** Cyan arrows show all action candidates considered, and orange arrows show 10 actions selected during an execution. We aid the baseline by limiting the candidates to the most effective actions (horizontal pushes for vines, tangential pushes for Dracaena).

**Figure 5: Hand-crafted dynamics model** that represents vines as vertically hanging spaghetti (left), and the Dracaena plant as 2D radially emanating spaghetti (right). Cyan area represents the space revealed by the action (red arrow) under this hand-crafted model.

**Generalization across Plant Growth Performance.** To test generalization, we run the single action selection experiment (Table 2) on the Dracaena plant after two-month of growth (in August) but with a model trained on data from June. Note the difference between plants in Figure 6 (top). Despite changes in appearance and leaf length, PPG generalizes to the grown Dracaena and outperforms random tangential action as shown in Figure 6 (bottom).

**Long-horizon Exploration Performance.** Next, we study if SRPNet can be used for situations that require multiple sequential interactions to reveal space behind plants. The task is to maximize the cumulative space revealed over a 10 time-step episode.

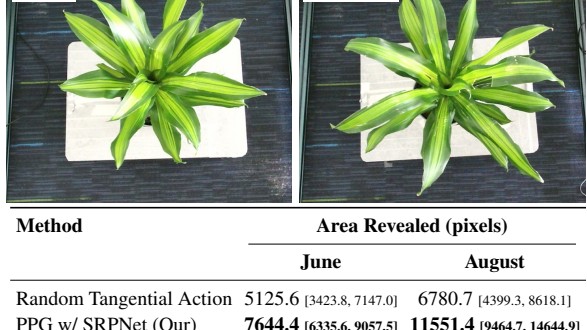

| Method | Area Revealed (pixels) | |
|---|---|---|
| | **June** | **August** |
| Random Tangential Action | 5125.6 [3423.8, 7147.0] | 6780.7 [4399.3, 8618.1] |
| PPG w/ SRPNet (Our) | **7644.4** [6335.6, 9057.5] | **11551.4** [9464.7, 14644.9] |

**Figure 6:** We evaluate a model trained with data collected from the June Dracaena on the August Dracaena.

This further tests the quality of SRPNet which now also needs to accurately predict *where* it thinks space will be revealed.

We conduct 4 experiments, one on Dracaena and 3 on vines. For the vines, we considered 3 settings: a) Base Setting: vine setting as used for collecting training data, and 2 novel settings to test generalization: b) Sparse Vines, and c) Separated Vines. While the last two settings explicitly test generalization, we note that the first setting also tests models on novel vine configurations not exactly seen in training. For Dracaena, we only conducted experiments in the Base Setting.

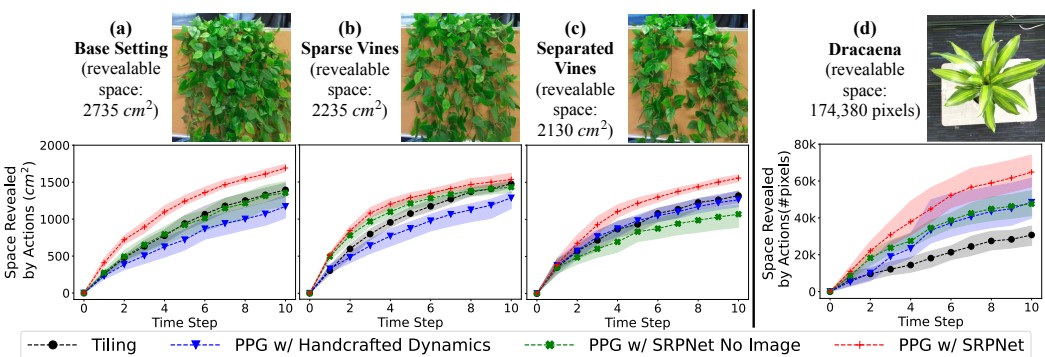

**Figure 7: Comparison of different methods for multi-step exploration of space behind plant foliage.** We show results in four settings across vines (left) and Dracaena (right). The line plots show average cumulative space revealed by actions up to time step $t$ across 10 trials (along with 95% confidence intervals). SRPNet training data was collected in the base setting shown in (a) for vines and (d) for Dracaena. (b) and (c) are novel settings that test generalization capabilities of our model. Our method (PPG w/ SRPNet) outperforms all baselines (a strong hand-crafted policy, and PPG with other dynamics models) across all settings.

Figure 7 plots the average space revealed (in cm$^2$ for vines and in pixels for Dracaena) as a function of the number of time-steps. We report the mean over 10 trials and also show the 95% confidence interval. Across all three experiments our proposed method achieves the strongest performance. Supplementary Figure S4 and videos on the website show some sample executions.

Results suggests that SRPNet is quite effective at predicting where space will get revealed (PPG w/ SRPNet *vs.* Tiling). Learning and planning via CEM lets us model complex behavior which is hard to hand-craft. Improvements over the tiling baseline increase as the action space becomes larger (Dracaena *vs.* vines). Moreover, benefits don't just come because of keeping track of revealed space ($C_t$), but also from the use of SRPNet (PPG w/ SRPNet *vs.* PPG w/ Handcrafted Dynamics). Furthermore, our model is able to interpret the nuances depicted in the visual information to predict good actions (PPG w/ SRPNet *vs.* PPG w/ SRPNet No Image). SRPNet also leads to benefits in novel vine configurations. Benefits are larger in the separated vines case than for the sparse vines. This may be because the separated vines are still locally dense and SRPNet processes local patches.

**Targeted Revealing Performance.** Our final experiment tackles the task of targeted exploration. The task here is to reveal space at a user-defined region, $m$. Figure 8 (left) shows a sample user-selected region. We tackle this task by setting $C_0$ to be $\bar{m}$, the complement of the user-defined region. Figure 8 (right) presents the results (same legend as for Figure 7, but without Tiling Baseline). Again, as SRPNet reliably models the effect of actions, PPG with SRPNet outperforms PPG with other dynamics models.

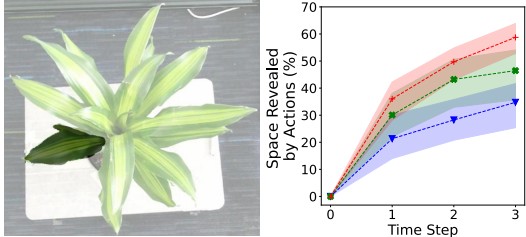

**Figure 8:** PPG is also effective at revealing space behind a specific spatial target.

## 6   Discussion

In this paper, we introduced PPG and SRPNet to tackle the problem of manipulating the external plant foliage to look within the plant (the *plant-self occlusion problem*). SRPNet uses self-supervised learning to model what space is revealed upon execution of different actions on plants. This sidesteps the difficulty in perception arising from dense foliage, thin structures, and partial information. PPG derives control strategies using SRPNet via CEM, to output sequence of actions that can incrementally explore space occluded by plants. Experiments on a physical platform demonstrate the benefits of our proposed learning framework for tackling the plant self-occlusion problem.

## 7   Limitations

We believe ours is a unique and first-of-its-kind study, but it has its limitations. We note two failure modes. First, PPG sometimes resamples an overly optimistic action (that doesn't actually reveal much space, so nothing changes and CEM returns a very similar next action) many times over without making progress. Second, as each individual push action doesn't use visual feedback it can't recover from say when a leaf slips from below the gripper. These may be mitigated by incorporating spatial diversity while selecting actions and by learning closed loop leaf manipulation policies through imitation. More generally, our overall approach relies on input from RGB-D cameras that are known to perform poorly in the wild. This may be mitigated through use of specialized stereo cameras built for farm settings [56]. Our techniques for automatic estimation of revealed space can be improved further using recent point tracking models [57], and it may be useful to build models that can predict and keep track of full 3D space. Experiments should be conducted with more diverse real plants. Future work should also rank actions from the perspective of the damage they cause to the plant, perhaps via some tactile sensing [30]. Lastly, while autonomous agriculture provides a path towards sustainable agricultural practices, societal impact of such automation should be studied before deployment.

**Acknowledgments**

Images in Figure 1 (top) have been taken from YouTube video 1 and YouTube video 2. This material is based upon work supported by the USDA/NSF AIFARMS National AI Institute (USDA #2020-

67021-32799), an NSF CAREER Award (IIS-2143873), an Amazon Research Award, and an NVidia Academic Hardware Grant. We thank Matthew Chang and Aditya Prakash for helpful feedback. We thank Kevin Zhang for help setting up robot experiments.

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

# Push Past Green: Learning to Look Behind Plant Foliage by Moving It
## Supplementary Material

## A  Implementation Details for Vine Experiments

### A.1  Robot Action Space

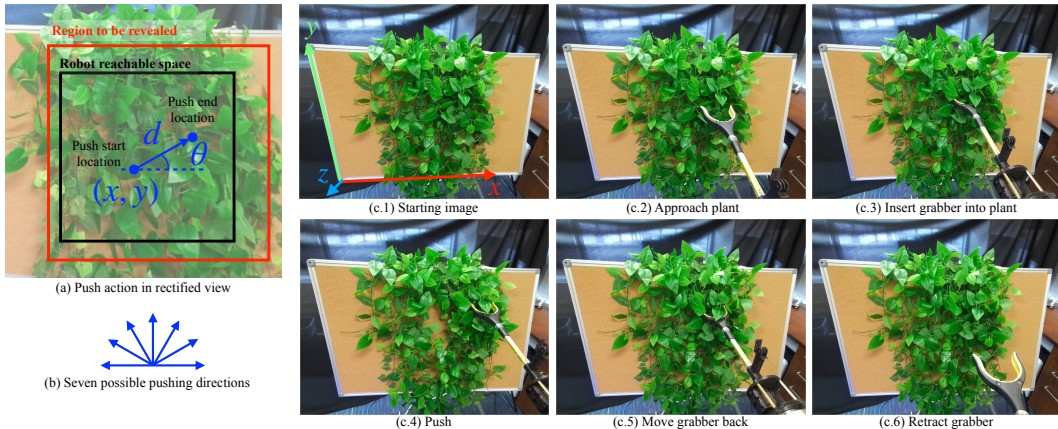

**Figure S1: Robot's action space for vine setup. (a)** shows the rectified image that we operate in, the region to be revealed (red box), and the region that the robot can reach (black box). The robot can execute push actions that start at a pixel $(x, y)$ in the rectified image and push a distance of $d$ at an angle $\theta$. We use 7 discrete push directions $\{0, \pi/6, \pi/3, \pi/2, \dots, \pi\}$ as shown in **(b)**. **(c.1) through (c.6)** show a sample execution of the push action.

The robot's action space consists of non-prehensile pushing actions. As shown in Figure S1 (a), these actions are parameterized by $(x, y, \theta, d)$. Such parameterization for pushing actions has been used in past works, *e.g.* [51]. Here, $(x, y)$ denotes the start location for the push interaction on the board, $\theta$ denotes the push angle, and $d$ denotes the push length. As shown in Figure S1 (b), we sample $\theta$ to be one of 7 angles from $\{0, \pi/6, \pi/3, \pi/2, 2\pi/3, 5\pi/6, \pi\}$. We do not sample angles greater than $\pi$ because pushing towards the bottom of the vines only drags down the vines and could pull the board over. We assume that the grabber inserts deep enough into the vines to push the vines but not too far to knock it over; therefore, the pushes are planar actions executed with the same $z$ value. We estimate the location and orientation of the board and establish a coordinate frame that is aligned with the board. Push locations and orientations are expressed in this coordinate frame. We implement these actions by moving the grabber through 4 waypoints, as shown in Figure S1 (c.2) to Figure S1 (c.5). In Figure S1 (c.4), we can see the effect of a randomly sampled action on the state of the vines. We drive the Franka Emika robot between these waypoints using the Franka-interface and frankapy library [58].

### A.2  SRPNet

For the vine setup, we are unable to position the camera such that it is perpendicular to the board. Therefore, we design SRPNet to work on rectified images of the scene, such that the camera is looking straight at the vines. This corresponds to using a homography to transform the image such that the surface underneath the vines becomes fronto-parallel. We build the model to only reason about a 40cm × 40cm neighborhood around the action start location. Parts of the board get occluded behind the robot arm as the robot executes the action. These occluded parts and area with no depth readings are masked out for evaluation and training.

### A.3  Data Collection

The robot's actions are in the same fronto-parallel plane used for SRPNet as described earlier. We estimate the space that can be safely reached by the robot ahead of time to make sure it is not close to its joint limits during interactions. The resulting space is roughly 40cm × 40cm. We divide the feasible space into a 20 × 20 grid. Action starting locations $(x, y)$ are sampled at the centers of these

| Push Angle | 0 | $\pi/6$ | $\pi/3$ | $\pi/2$ | $2\pi/3$ | $5\pi/6$ | $\pi$ | Full Dataset |
|---|---|---|---|---|---|---|---|---|
| **# Interactions** | 985 | 460 | 360 | 348 | 359 | 433 | 584 | 3529 |
| **Mean area revealed (cm$^2$)** | 215.7 | 177.3 | 93.6 | 58.8 | 100.4 | 180.9 | 237.1 | 170.3 |

**Table S1: Statistics for the different push directions in the collected vine dataset.** Collected dataset reveals many aspects of the problem. For example, for vines, horizontal push actions (0 and $\pi$) are the most effective at this task.

grid squares (*i.e.*, 400 possible starting locations). We sample push directions from the 7 possible angles, $\{0, \pi/6, 2\pi/6, \ldots, 6\pi/6\}$, and push by 15cm clipping to the feasible space as necessary. Therefore, not all interactions have $d = 15$; for starting locations near the boundary, $d < 15$.

Our full dataset contains 3529 interactions (summed to roughly 30 hours) collected over 11 different days (nonconsecutive). This data includes 2571 interactions done specifically for the purpose of data collection. The remaining interactions come from when we were developing control algorithms. These don't follow uniform sampling from the robot's action space and are biased towards horizontal actions since the most effective actions for the baselines are often horizontal actions.

We automatically compute the ground truth for training the model on the collected data. Specifically, we use color thresholding to determine when the surface beneath the vines has been fully exposed. We found this simple strategy to be reasonably robust. Note that while we train and use SRPNet to predict whether *all* vines were moved aside to reveal the board, we can process the data in other ways to also train the model for other tasks. For example, we can re-purpose the data for a task that involves only looking beneath the first layer of vines. We can re-compute ground truth to identify locations where the height decreased by (say) more than 5cm for such a task.

## A.4 Cross-entropy Method

Our CEM implementation uses 3 iterations that each evaluate 300 candidate actions. We sample $(x, y, \theta)$ from Gaussian distributions. In the first CEM iteration, $x, y, \theta$ are sampled from Gaussians with different mean and variances, chosen to cover the whole action space. The parameters are then discretized to match the distribution from data collection. When sampling actions, we only retain action samples that are feasible (*i.e.* within the robot's reachable space as shown in Figure S1 (a)). Elite samples are the top $20\%$ candidates that have the most amount of *new space* revealed. Running line 3 to 6 in Algorithm 1 (Section 4.3) for vines takes about 5 seconds.

# B  Implementation Details for Dracaena Experiments

## B.1  Robot Action Space

The robot's action space for Dracaena is similar to that of vines. However, since the Dracaena leaves are at different heights, we define three possible $z$ values that the grabber can insert to. The Dracaena plant body is about 45cm tall so we defined the $z$ values to be about 22.5, 17.5, and 12.5cm from the top of the plant. For each $z$ value, planar pushing actions $(x, y, \theta, d)$ are defined on a plane parallel to the ground. We sample $\theta$ from 8 possible angles: $\{0, \pi/4, \pi/2, 3\pi/4, \pi, 5\pi/4, 3\pi/2, 7\pi/4\}$. The angles are 45 degrees away from one another instead of 30 degrees as used in vines because we want to keep the total number of possible actions reasonable.

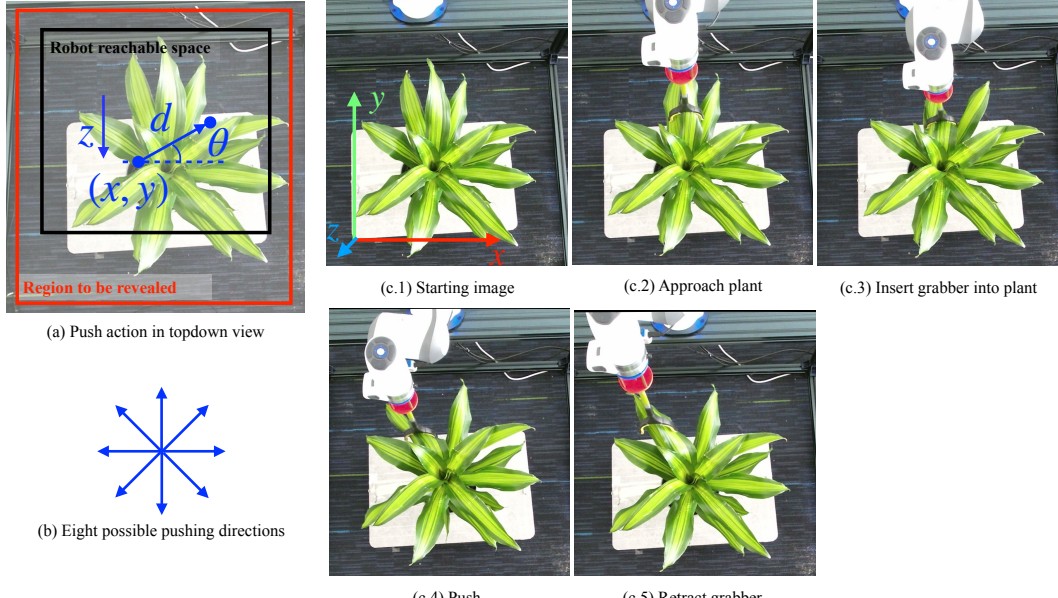

(a) Push action in topdown view

(b) Eight possible pushing directions

(c.1) Starting image   (c.2) Approach plant   (c.3) Insert grabber into plant

(c.4) Push   (c.5) Retract grabber

**Figure S2: Dracaena robot action space.** Similar to Figure S1, **(a)** shows the image from the camera, **(b)** shows the pushing directions, and **(c)** shows the sample execution of a push action.

## B.2  SRPNet

Since the Kinect camera is looking down at the Dracaena plant, SRPNet does not work on rectified images as it does for vines and instead takes in images from the camera as they are. We project action start locations into their image coordinates using the camera intrinsics and crop around the locations to obtain local patches to input into the network. When training SRPNet, adding another head to predict height decrease in addition to the binary classification head helps AP performance. We use Huber loss with $\delta = 0.1$ to provide an auxiliary loss to the network.

## B.3  Data Collection

The reachable space of the robot in the Dracaena setup is roughly $57\text{cm} \times 53\text{cm}$ and corresponds to a $29 \times 27$ grid of 2cm cells. Similar to the vines' setup, the action starting point $(x, y)$ is sampled from these 783 possible locations. Given that pushing from the center of the plant tends to displace it entirely, we aim to discourage such actions to prevent damage to areas where new leaves may sprout. We manually delineate a rectangular region around the plant center and do not sample or execute actions in this region. We also sample $z$ from 3 possible values (22.5, 17.5, and 12.5cm from the top of the plant as mentioned before), push directions from 8 possible angles, $\{0, \pi/4, \pi/2, 3\pi/4, \pi, 5\pi/4, 3\pi/2, 7\pi/4\}$, and push by 15cm clipping to the feasible space as necessary. Therefore, not all interactions have $d = 15$; for starting locations near the boundary, $d < 15$.

Since the plant wobbles during pushing, we discount the area that is revealed due to whole-plant movement. We construct plant point clouds before and after an action; then, iterative closest point (ICP) is performed to align the two point clouds. During execution, the robot body occludes parts

| Push Angle | 0 | $\pi/4$ | $\pi/2$ | $3\pi/4$ | $\pi$ | $5\pi/4$ | $3\pi/2$ | $7\pi/4$ | Full Dataset |
|---|---|---|---|---|---|---|---|---|---|
| # Interactions | 257 | 262 | 295 | 273 | 249 | 297 | 289 | 253 | 2175 |
| Mean area revealed (pixels) | 1391.4 | 1138.7 | 990.4 | 802.0 | 1154.0 | 1110.8 | 1154.9 | 1495.2 | 1147.7 |

**Table S2: Statistics for the different push directions in the collected Dracaena dataset.**

of the plant, so we mount a Intel RealSense camera at the wrist to fill in these occluded regions to aid ICP. Area where the plant height has decreased in the aligned point cloud is considered to be revealed space.

## B.4    Cross-entropy Method

We follow the same algorithm as the one outlined in Algorithm 1 (Section 4.3). The Dracaena CEM uses 3 iterations that each evaluate 300 candidate actions. We sample $(x, y, \theta, z)$ from uniform distributions within the robot's reachable space. The parameters are then discretized to match the data collection's distribution. Top 20% candidates that reveal the most amount of new space are chosen as elite samples that are fitted with Gaussian distributions for the next iteration. Running one iteration takes about 7 seconds.

## B.5    Comparing Tangential to Random Actions

| Method | Area revealed (pixels) |
|---|---|
| Random Action | 3956.1 [2826.1, 5045.9] |
| Tangential Action | **5125.6** [3423.8, 7147.0] |

**Table S3: Effectiveness of tangential actions.** We execute actions tangent to Dracaena leaves in the Tiling baseline because they reveal more space on average compare to random actions.

We chose horizontal actions for the Tiling baseline of vines because they on average reveal the most amount of space. In order to come up with a similar Tiling baseline for Dracaena, we observe that leaves are pushed aside more easily when the grabber moves tangent to the leaves. We verify that tangent actions are better than random actions by comparing average space revealed upon execution of actions from the two methods. As shown in Table S3, tangential actions reveal more space than random actions, so we use them in the Tiling baseline to test the effectiveness of PPG w/ SRPNet against this strong baseline.

# C   Visualizations

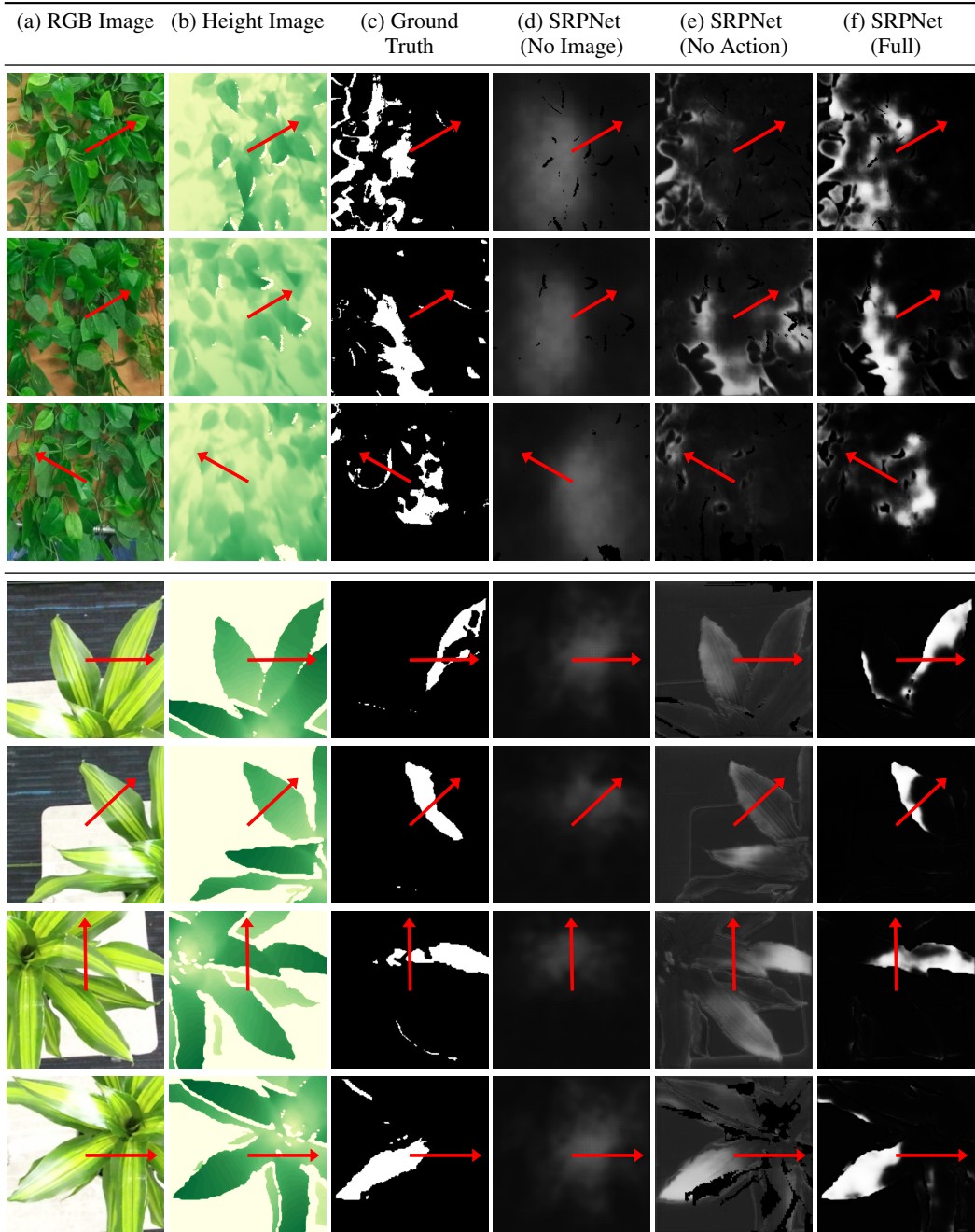

|  (a) RGB Image | (b) Height Image | (c) Ground Truth | (d) SRPNet (No Image) | (e) SRPNet (No Action) | (f) SRPNet (Full) |

**Figure S3: Visualizations of output from our proposed SRPNet.** We show examples from the test set. The white regions in ground truth images represent space revealed by actions drawn as red arrows. Column (d) shows prediction from SRPNet without image input (*i.e.* no RGB, no height), column (e) shows prediction from SRPNet without action input, and column (f) shows predictions from SRPNet. The brighter the region, the higher the predicted probability of revealing space. Ground truth revealed space indicates the complexity of the task and suggests why the hand-crafted dynamics model (shown in Figure 5) performs poorly at this task. SRPNet is able to effectively use the visual information to make good predictions.

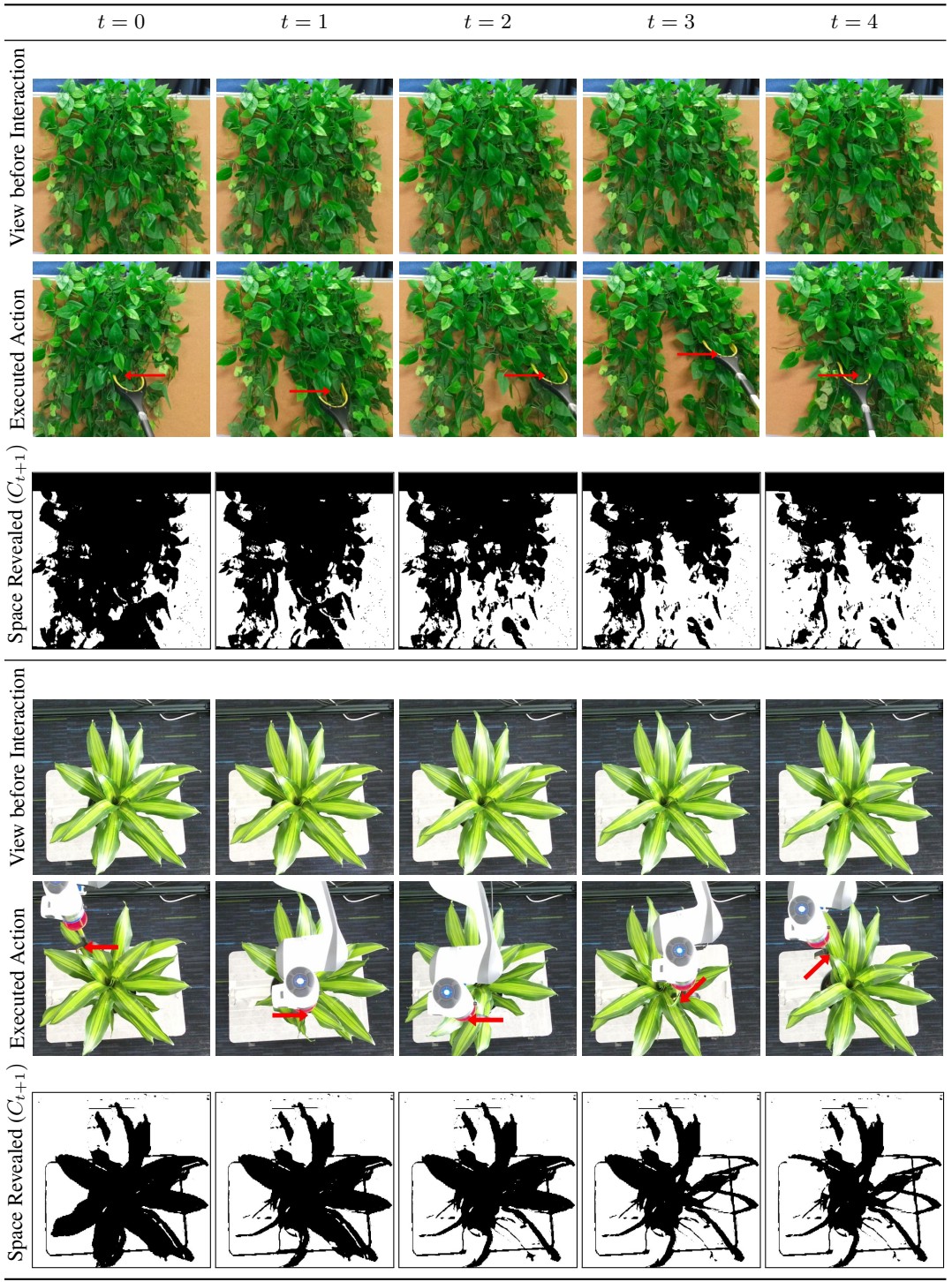

**Figure S4: First five time steps of a sample execution from our method.** Top row shows the RGB image before interaction, middle row shows the push action executed, and the bottom row shows the cumulative space revealed so far. Our model picks actions that are effective at revealing space.

