# OpenReview forum: "Push Past Green: Learning to Look Behind Plant Foliage by Moving It"
_robot-learning.org/CoRL/2023/Conference — CoRL 2023 Poster_

### Official Review · Reviewer_N9Cs · 2023-06-27

**Confidence:** 4
**Originality:** Very Good
**Technical Quality:** Very Good
**Clarity Of Presentation:** Good
**Impact:** 4

**Recommendation:**

Strong Accept: I recommend accepting the paper and will argue for my recommendation even if other reviewers hold a different opinion.

**Review:**

**STRENGTHS:**

* **Highly relevant from a robotics-in-agriculture perspective**, and this relevance is well motivated in the main text (e.g. the introduction).

* The suggested approach is well motivated, e.g. Line 29-38 in the introduction. In particular, it motivates the need for the collected dataset. Also, to the best of my knowledge (I'm not an expert on robotics-in-agriculture), the work is put into context of relevant related work (see the related works section) and the suggested approach is well-motivated given the various shortcomings / missed opportunities of past works. To the best of my knowledge, the approach and type of dataset are novel.

* A **relevant, real-data and large-scale dataset has been collected** and will be made **publicly available for future research** within the plant self-occlusion problem.

* **Great that the method can be (and is!) trained in a self-supervised manner**, so that expensive annotations can be avoided.

* **Strengths related to the experimental results:**
    - **Good set of ablations** that showcase the design choices (see e.g. Table 1). The associated analyses (see e.g. Line 199-211) are good too, and provide interesting insights (e.g. "There are no trivial solutions of the form 'space gets revealed where height is high'.").
    - Good that **multiple runs with different random seeds are done**, and that confidence intervals are included.
    - Good that **runtimes are reported** (albeit in the supplementary material; such an important metric may be best to add to the main paper).
    - The **proposed method outperforms reasonable baselines.**

* **Very impressive additional material**, including the video summary on the attached webpage.

**WEAKNESSES:**

* **Some things were not quite clear to me and would improve the paper if addressed:**
    - **Why the use of a CRM approach for action selection?** It would be good to see some experimental results that compare some form of pure argmax-style action selection during evaluation.
    - In Table 2, I interpreted the "Random Horizontal / Tangential Action" as equal to the "Tiling Baseline". If this is true, consider renaming "Random Horizontal / Tangential Action" to "Tiling Baseline". If not, consider clarifying.
    - It's sometimes hard to grasp the Area Revealed metric when it is reported e.g. in cm^2. Could one perhaps also include some metric which is in the [0,1]-range, i.e. normalized somehow?
    - It was a bit difficult for me to understand the difference / connection between "PPG w/ Other Dynamics Models" and "Hand-crafted Dynamics Model". It seems like the latter could be a subset of the former (?).
    - In Fig. 7, I was a bit confused why the methods require multiple steps to reveal a single plant part? It seems a single, deterministic step would do it, given that the targeted area is quite small (?).

* The greedy nature of the reveal actions imply that **the proposed methodology may be inherently ill-suited for certain types of inferences.** The greedy reveal actions are certainly good for the specific task of revealing things (assuming all reveals are considered equally important / informative -- see my next remark in the below item), but if the revealing is merely a step towards some form of 'global-level' inference related to the plant (e.g. its overall health status), then greedy reveals may not be the globally optimal approach for said global inference. An RL-based method with a sparse reward (obtained at the end) could be a better choice in such cases, but on the other hand it is well-argued why such an approach is difficult in these contexts (see e.g. the second paragraph of the intro).

* Oftentimes, **not all visible space is equally important for a given type of inference** (it may e.g. be that some parts of a plant typically does not hide anything 'interesting', and so more focus should be put on revealing those parts that are likely to hide something interesting (e.g. berries)). It would have been great to see some discussion (e.g. in the Limitations section) that mentions this fact, and ideas for addressing it.

* In the same spirit as the last item above: **It would have been great to have some form of experimental results on downstream performance** (e.g. the task of counting things that are occluded by plant parts (e.g. berries)), and not only evaluate the "reveal" metric for which the model is explicitly trained. I however understand that the scope of a paper has its limits.

* While there is quite some degree of out-of-domain generalization being showcased, some quite simple tricks could have further assessed this. For example, the data collection could have been done in different lightning conditions, so that models trained in a certain lightning condition would be tested on another condition.

* Again, great with seed sensitivity analyses. But since the number of trials is less than 20 (it's 10 as I understood it), perhaps +- std would have been sufficient, rather than 95% intervals (since they are not statistically significant given the few trials (less than 20), if I don't misremember my statistics-101-class).

**SUGGESTIONS (not part of evaluation):**

* Might be nice with a last column in Table 1, that simply computes the average numbers across the three existing columns.

* Might be interesting future work to combine your proposed methodology with some form of methods that predict what lies beneath plant parts (similar to e.g. human pose estimation, where occluded body parts are often estimated, albeit at varying accuracy). The policy may then adapt is "reveal movements" based on how accurate it thinks that the model already is at predicting what is beneath the plant part even before the reveal. In other words, only those plant parts are revealed where the model cannot confidently guess what is beneath.

**Quality Of The Limitations Section:**

Limitations are addressed clearly

**Questions For Rebuttal:**

Please see my comments / questions under "WEAKNESSES" above. Prioritize answering any comments which include a question, e.g. "Why the use of a CRM approach for action selection? It would be good to see some experimental results that compare some form of pure argmax-style action selection during evaluation."

If space permits, consider also addressing any other comments I have.

**Robotics Focus:**

Sufficient demonstration on hardware

**Summary Of Paper:**

This paper presents a self-supervision-based approach -- Push Past Green (PPG) -- for learning to de-occlude parts of a plant by sequentially moving plant parts with a (real) robot arm. It also introduces a novel dataset for two different plant types (vines and Dracaena) that consists of ~50 hours of plant interaction and associated ground truth "space revealed" metrics (data collected using random robot arm actions). This dataset is used to train the PPG system via self-supervision on the random actions in the dataset, and associated "space revealed" scores.

The model input is an RGB + height image of the plant, a history of space revealed so far, as well as an encoding of the suggested action. A subnetwork called "Space Revealed Prediction Network" (SRPNet) takes the action encoding and a crop around the start pixel, encodes this and decodes it into a probability map of space revealed.

Extensive experimental results (using a real robot) showcase the advantages of the method.

**Summary Of Recommendation:**

The paper is very well-written, proposes a solid approach and dataset for an important task within robo-for-agriculture, and extensively showcases the merits of the proposed method and its various design choices. I think the paper should definitely be accepted. It may however be further improved in various ways (see my comments above), but all-in-all it's a no-brainer for acceptance in my view.

---

> ### Author Response · Authors · 2023-08-11
> **Response to questions**
>
> Thank you for the encouraging feedback and the thoughtful suggestions. We respond to your specific questions below.
>
> 1. **Why the use of a CEM approach for action selection?** The action space is large (29x27x3x8 = 18K actions for Dracaena), brute force search will take 5 minutes on a 2080 GPU. Our CEM implementation only evaluates 900 total actions over 3 iterations and thus speeds up the search for high-performing actions (21 seconds).
>
>     We also conducted an experiment to compare CEM against argmax for action selection. We use our original SRPNet model (without any re-training) on the same Dracaena plant that has grown and changed during the past 2 months. (See the plant difference in https://sites.google.com/view/pushingfoliage/home?authuser=2#h.3szqmxp39t7n ). We observe the following space-revealing performance for selecting a single action:
>
>     | Method |  mean and 95% CI | Action Selection Time (s) |
>     | -------- | -------------| ------------- |
>     | Random Tangential     | 6780.7 [4399.3, 8618.1]   | 0 |
>     | Argmax for action selection| 11110.8 [9294.7, 13687.7] | 300 |
>     | CEM for action selection(Our)  | 11551.4 [9464.7, 14644.9]   | 21 |
>
>      Note that the absolute numbers are different from ones in Table 2 because the plant has grown. Argmax and CEM pick actions that give comparable results, but CEM is 15x faster at outputting actions.
>
> 2. **"Random Horizontal / Tangential Action" same as "Tiling Baseline"?** Yes, they are the same. "Tiling Baseline" is essentially executing multiple "Random Horizontal (for vines) / Tangential Actions (for Dracaena)" in a spread-out manner for long-term exploration. In Table 2, we only execute one action thus we used text that directly described the action. We will clarify in the paper.
>
> 3. **Include normalized area metric.** Thank you for the suggestion on normalizing the reported metric. We include in Figure 6 how much revealable space there is for each setting. (For example, Sparse Vines has 2235 ${cm}^2$.) We will consider updating the figures with an additional normalized axis.
>
> 4. **"PPG w/ Other Dynamics Models" vs. "Hand-crafted Dynamics Model"** Yes, "Hand-crafted Dynamics Model" is a subset of "PPG w/ Other Dynamics Models", since the hand-crafted models can be plugged into PPG to replace SRPNet. We will alter the section headings to make it clearer.
>
> 5. **Multiple steps for targeted reveal?** Due to how leaves are attached to the center of the plant, the part of the leaf near the center moves very little, so it is difficult to reveal 100% of the leaf. The model predictions also have errors, so it cannot push away everything in the first try or leaves might slip.
>
> 6. **Disadvantage of greedy approach / application to downstream task** We agree that there may be many applications that require looking at specific places in a plant (e.g. where berries are likely to be found, or where rot is likely to occur). One way could be to decompose into a) predicting “fruitful” locations to explore, and b) how to move foliage to expose space at those locations. We admit that we don’t tackle the former. But, we believe our methodology provides a possible answer for the latter (such as our Targeted Revealing experiment on L276).  For the former, we need to build a model to predict where targets could be. RL may be suitable to build such an task-specific expectation and PPG w/ SRPNet could serve as an performant action primitive to speed up learning or as exploratory actions. Imitation learning, or even human labeling in images could be other alternatives. Thank you for this useful feedback. We will add discussion on this in the limitations section.
>
> 7. **Generalization / Lighting conditions** Our data collection and experiments are carried under varying natural light. So it is hard for us to assess such generalization systematically (for example, certain afternoons could be cloudy making it seem like a morning). We will keep the suggestions in mind for the future. The argmax experiment (point 1 above) was conducted on Dracaena that has grown over the last 2 months. Our model still performs well. This further tests generalization.
>
> 8. **Number of trials and confidence interval** We agree that more trials will tighten the bounds further. Our current implementation uses `scipy.stats.t.interval`, and it assumes normality of the distribution. Since we only have 10 samples, it would be more appropriate to use `scipy.stats.bootstrap`. This change does not affect observations and conclusions we state in the paper, since the two bounds are very similar. New figures and tables (along with the old ones for comparison) can be found in: https://sites.google.com/view/pushingfoliage/home?authuser=1#h.98is7ovedkrd
>
> Hopefully we have addressed the concerns raised in this review. We will love to hear your feedback, and we are happy to offer further clarifications or respond to any other concerns. We hope our response helps improve the impression of our work.

---

> > ### Comment · Reviewer_N9Cs · 2023-08-11
> > **Satisfied with rebuttal**
> >
> > This post is just to acknowledge that I've carefully read the rebuttal (both the rebuttal to my own review, but also that of the other reviewers). I believe the rebuttals were satisfactory overall, and my initial judgement 'strong accept' stays firm. To re-state my original recommendation:
> > "The paper is very well-written, proposes a solid approach and dataset for an important task within robo-for-agriculture, and extensively showcases the merits of the proposed method and its various design choices. I think the paper should definitely be accepted. It may however be further improved in various ways (see my comments above), but all-in-all it's a no-brainer for acceptance in my view." <-- and those various small things were well-addressed in the rebuttal.

---

### Official Review · Reviewer_qgE2 · 2023-07-20

**Confidence:** 3
**Originality:** Good
**Technical Quality:** Good
**Clarity Of Presentation:** Good
**Impact:** 2

**Recommendation:**

Weak Accept: I recommend accepting the paper, but will not argue for my recommendation if the majority of other reviewers have a different opinion.

**Review:**

### strengths
1. Autonomous agriculture is an important but less-explored area. This paper proposed a data-driven method to solve an important challenge in this field.
2. The proposed method is straightforward and easy to understand. The cross-entropy method, simple yet effective, enables multi-step closed-loop control.
3. The proposed method and comprehensive ablations are evaluated on plants in the real world.

### weaknesses
1. Action space is not clearly described. For example, how many bins are there for one-hot orientation and z height. Related to this, what's the reason to use one-hot representation for orientation and z height, instead of absolute value?
2. The task solved in this paper is to maximize the cumulated revealed space. Why this task is important and how does the proposed method help other downstream tasks?
3. The proposed method is only evaluated on two specific plants. It would be better to conduct a systematic and slightly large-scale evaluation.
4. plants are deformable and elastoplastic objects. The proposed method doesn't explicitly model the physical properties and dynamics.

**Quality Of The Limitations Section:**

Limitations are addressed clearly

**Questions For Rebuttal:**

refer to the weakness part in the previous section.

**Robotics Focus:**

Sufficient demonstration on hardware

**Summary Of Paper:**

This paper focuses on an autonomous agriculture application. It proposed a learning-based method (SRPNet) to manipulate the plant foliage to look behind the leaves and the branches. The proposed SRPNet predicts revealed space for each action candidate and uses the cross-entropy method to achieve closed-loop control. Finally, the proposed method is evaluated on two plants in the real world.

**Summary Of Recommendation:**

New application for robot learning and comprehensive evaluation and ablation study on real-world platforms.

---

> ### Author Response · Authors · 2023-08-11
> **Response to questions**
>
> Thank you for the encouraging feedback and the thoughtful suggestions. We respond to your specific questions below.
>
> 1. Details about the robot action space are in Section A.1 for vines and Section B.1 for Dracaena in the supplementary. We summarize it briefly here. For vines, in Figure S1 (b), we sample $\theta$ to be one of 7 angles from {0, $\pi$/6, $\pi$/3, $\pi$/2, 2$\pi$/3, 5$\pi$/6, $\pi$} and use a constant z as vines' foliage lies roughly on the same surface. For Dracaena, we sample $\theta$ from 8 possible angles: {0, $\pi$/4, $\pi$/2, $\ldots$, 7$\pi$/4}. We also sample z from 3 possible values (22.5, 17.5, and 12.5cm from the top of the plant). We choose these z values so that leaves at different levels can be reached.
>
>     We use one-hot representation, because the angle and z height have discrete values. Past works similarly use one-hot representations when dealing with discrete valued inputs (e.g. discrete actions, task IDs) as network input [DensePhyNet, MetaWorld]. We conduct experiments to see if taking in raw absolute values would make a difference. In the same settings as Table 1:
>
>     | Methods      | Vines [All] | Vines [5cm] | Dracaena | Mean |
>     | ----------- | ----------- |  -----------  | -----------  |----- |
>     | Full SRPNet (Our, one-hot)      | 46.3   | 54.4 |44.2    | 48.3 |
>     | Full SRPNet (raw absolute value)   | 47.8 | 55.1  | 38.6        | 47.2 |
>
>     While use of raw values improves performance a little bit for vines ( $\leq 1.5$% increase), it degrades performance quite a bit for Dracaena (a 5.6\% drop). On average, the one-hot representation works better.
>
> 2. Plant self-occlusion is common. Even as humans, we routinely push around external foliage for inspection or reaching into the plant. See Figure 1 (left) or the original videos from which these frames were taken (https://www.youtube.com/watch?v=M9eT4DJLUB4&t=532s and https://www.youtube.com/watch?v=Oa2apuOj5R4&t=984s). Thus, it is desirable to equip robots with a similar capability. At the same time, manipulating plants presents its own unique challenges different from manipulating other deformable objects (e.g. lack of simulators, more structure).
>
>     We believe solutions to this plant self-occlusion problem will have direct applications in plant inspection and reaching within a plant. It can be combined with existing agricultural systems [10, 19] for say tasks like fruit harvesting [18, 21] which may require exploring behind a given region in a plant. The “Targeted Revealing” experiment on L276 explicitly tests the ability to look behind specific parts of a plant, and we found our proposed model to be effective at it.
>
> 3. We agree and note in the limitations sections that a larger and broader study should be done. However, we believe that our current study is quite systematic and already reveals important aspects about the problem (e.g. insufficiency of hand-crafted heuristics, need for data-driven methods). Techniques that scale-up learning and evaluation of such policies to the diverse plants around us will be future work.
>
>     As a step towards an expanded evaluation, we have conducted an experiment that tests the robustness of our models to changes in the plant as it grows. We repeat the one-step experiment (from Table 2 in the paper) on the Dracaena two months after, without collecting any data on the changed plant. See the difference in plant over these 2 months at https://sites.google.com/view/pushingfoliage/home?authuser=2#h.3szqmxp39t7n . We find that our models generalize reasonably to such changes induced by growth and our method is still able to select an effective revealing action:
>
>     | Method |  mean and 95% CI |
>     |--------|-------------|
>     | Random Tangential     | 6780.7 [4399.3, 8618.1]   |
>     | PPG w/ SRPNet (Our)  | 11551.4 [9464.7, 14644.9]   |
>
> 4. In some way, we are implicitly modeling plant dynamics, to the extent necessary, for our downstream task. For SRPNet to accurately predict  where space would get revealed after an action, it must reason about the effect of an action on the current state, e.g.whether the leaf will move with the gripper or will it slip.
>
>     At the same time, it is not a full dynamics model, it only models as much is needed to solve the task at hand. This has an advantage of being more sample efficient, but at the same time the learning is task-specific and may not be useful for another task. Building a general purpose dynamics model will be future work.
>
> Hopefully we have addressed the concerns raised in this review. We will love to hear your feedback, and we are happy to offer further clarifications or respond to any other concerns. We hope our response helps improve the impression of our work.
>
> **References:**
>
> [DensePhysNet]: DensePhysNet: Learning Dense Physical Object Representations via Multi-step Dynamic Interactions, RSS 2019.
>
> [Meta-World] Meta-World: A Benchmark and Evaluation for Multi-Task and Meta Reinforcement Learning, CoRL 2019.

---

### Official Review · Reviewer_23Je · 2023-07-20

**Confidence:** 4
**Originality:** Excellent
**Technical Quality:** Excellent
**Clarity Of Presentation:** Excellent
**Impact:** 4

**Recommendation:**

Strong Accept: I recommend accepting the paper and will argue for my recommendation even if other reviewers hold a different opinion.

**Review:**


### Strengths

- **S.1** Robotics is already hard and adding plants into the mix is a lot harder. I have huge respect for the authors choosing to tackle such a difficult (and "messy") problem and providing real-world experiments.
- **S.2** The method is clean, easy to follow, easy to implement, yet very useful.
- **S.3** The paper is well-structured and well-written. All the figures are very illustrative and a lot of attention went into everything. This reads like a very polished paper.
- **S.4** The experiments illustrate the strength of the contribution well.

### Weaknesses

I honestly can't think of any major concern. My only issue is that the authors use a 2D model for view novelty, not a 3D model but (a) that's not in the scope of this work and (b) the authors clearly state this as future work. The current submission is great as it is.
I was about to write that I would've liked a video to go along with the submission until I remembered that you referenced a website at the beginning of your paper. I checked out the videos on the website and they're very good.

**Quality Of The Limitations Section:**

Limitations are addressed clearly

**Questions For Rebuttal:**

I have a couple more nitpicks and questions, but nothing I'd reject a paper over:

- **Q.1** Less a question/nitpick and more of a recommendation: Maybe it'd be worthwhile investigating an eye-in-hand approach for the robot gripper, i.e. instead of having a static camera, you could build a live 3D reconstruction (e.g. a NeRF) by attaching the camera to the end effector and something like Nvidia's Instant-NGP (https://github.com/NVlabs/instant-ngp, which can create a NeRF in seconds on a recent GPU)
- **Q.2** You mentioned in your limitations that CEM sometimes retries the same action over and over. Shouldn't CEM update its parameters? I don't fully understand why it'd be doing that.
- **Q.3** I very much like your Fig. 4 and 5 and the captions for both - in my opinion, good figures are self-explanatory together with their captions and you're doing exactly that. The caption contains a very brief summary and explanation what we're seeing. Along the same "easy to follow" lines, I also greatly appreciate Fig.2 because you opted to include example images throughout the diagram that immediately make clear what's going on here without complex formalisms.
- **Q.4** So your model is pretty plant-type-specific at the moment, right? You train one model for each type of plant because they have very different leaf dynamics, right? Analogous to the spaghetti model, have you considered pre-training your model in simulation? It could be possible to create a large scrunched up sheet of cloth (or virtual spaghetti if you please) in a simulator that's capable of dealing with cloth (like Mujoco)
- **Q.5** In Tab.1, the performance of. the method without height map is better than with. Do you have any idea why that could be?
- **Q.6** How did you arrive at 15cm moving distance (line 147)? And how does this affect the risk of damaging the plant? In general, did you see any accidental injury to the plants?
- **Q.7** This method reminds me a little of DensePhysNet (https://arxiv.org/abs/1906.03853), where a model learns how certain dynamic actions will affect objects and it slowly builds up a model of the physics of every object in the scene. Have you ever considered making this a recurrent model (to track changes over time and explored space better) or would that not be useful in your scenario because sometimes the plant slips out of the end effector?

**Robotics Focus:**

Sufficient demonstration on hardware

**Summary Of Paper:**

The authors introduce a robotic method for investigating plants. A large part of that is moving foliage out of the way and the paper presents (a) a neural net that estimates how a certain manipulation would affect visibility and (b) a policy that successively explores the plant, trying to maximize viewing new, previously occluded areas over the course of several manipulations.

**Summary Of Recommendation:**

This is my favorite paper at this year's CoRL and I'll stand fight to get it accepted. It's very well-written, and presents a straight-forward solution and great experiments for a very difficult problem.

---

> ### Author Response · Authors · 2023-08-11
> **Response to questions**
>
> Thank you for the encouraging feedback and the thoughtful suggestions. We respond to your specific questions below.
>
> 1. Thank you so much for the suggestion on attaching the camera to the end effector and using Instant-NGP. For our next steps, we are considering how implicit representations can be used to learn better plant dynamics models.
>
> 2. SRPNet’s speculations about the space-revealing potential for an action can have errors. Specifically, if SRPNets drastically overestimates the new space an action can reveal, an action which is actually ineffective can  get selected as an elite. If CEM outputs such an action, after actual robot execution, not much additional space gets revealed, and  $C_{t+1}$ would be similar to $C_{t}$. At $t+1$, its inputs are all almost the same, so it again outputs a similar erroneously overoptimistic action. While CEM updates its parameters across iterations within a time step, it is re-initialized with uniform distribution at the next time step. In our current implementation, after a few time steps (say $k$), once $C_{t+k}$ differs sufficiently from $C_{t}$, then other actions are selected. Another term that encourages diverse actions to be sampled over the course of execution could better mitigate this issue.
>
> 3. Thanks for the encouraging feedback. We are glad that the figures helped to make things clear.
>
> 4. Yes, we train one model per plant. And, yes, it is because dynamics vary greatly across leaves and plants. For the real Dracaena plant, data collection and experiment spanned across days, and we let it grow and change.
>
>     Use of simulation is a promising and fruitful direction for future research. But it comes with its own challenges: simulating dynamics of individual leaves (e.g. leaves bending and slipping against the end-effector), setting up connectivity of leaves with stems, and modeling interactions of leaves with one another.
>
> 5. Good question, we don’t precisely know. It might be that the height map doesn’t add much over and above the coarse geometry directly inferable from the RGB image for the Vines [All] setting. We have also run training with 5 seeds and it looks like the 0.6 difference is within the noise:
>
>     | Method |  Vines[All] |
>     |--------|-------------|
>     | Full SRPNet (Our)      | 46.4 $\pm$ 0.5   |
>     | No height map  | 46.6 $\pm$  0.4   |
>
>     So, it's not quite that without height map is _better_ than with height map. Note that the other results in Table 1 all have a much larger gap.
>
> 6. If the pushing distance is too small, pushing has little discernible impact on the foliage. If the length is too large, the vines are strained, and we risk pulling it out. For Dracaena, pushing for too long also may twist or bend the leaves too much. Its leaf is stiff around the center, so if the grabber pushes far into the center, the leaves crack. We also note that the grabber itself is 10cm wide.
> We did notice some accidental injury during initial development and data collection: leaves getting torn and yellowing of leaf edges.
>
> 7. Thanks for the suggestion. We hadn’t thought of this and it will be interesting to explore. In our case, we explicitly maintain a map to keep track where space has been revealed ($C_t$ in Figure 2 and Algorithm 1) instead of a latent representation. Since the map is quite large, it might be difficult to store and update it within a RNN. However, similar to DensePhysNet, one could use a RNN to encode properties like stiffness or density for deformable objects, but even a single leaf can be stiff at different parts; it would be an interesting future work.
>
> Hopefully we have addressed the concerns raised in this review. We will love to hear your feedback, and we are happy to offer further clarifications or respond to any other concerns. We hope our response helps improve the impression of our work.

---

### Official Review · Reviewer_raTR · 2023-07-24

**Confidence:** 3
**Originality:** Good
**Technical Quality:** Fair
**Clarity Of Presentation:** Very Good
**Impact:** 2

**Recommendation:**

Weak Reject: I recommend rejecting the paper, but will not argue for my recommendation if the majority of other reviewers have a different opinion.

**Review:**

Strengths:

- The paper is mostly clearly written and easy to follow.
- This is an interesting application that has potential.
- SRPNet follows similar ideas as in other applications (depth completion, semantic scene completion, temporal forecasting, world models) where data-driven methods have shown good predictive results.

Weaknesses:
Three main concerns raised from this paper:
1. How to decide for an action it was not totally clear. From algorithm 1 it seems that every time there is a need to sample all possible action (used in training) and then evaluate SRPNet to then take the action that maximizes the information gain. This scales poorly with the action space and the complexity of the plant. Please comment.
2. The whole PPG idea reads like an ungrounded heuristic. There are plenty of methods for motion planning that could have been adapted to this application, giving the right observation and action spaces, for instance Next Best View (like to maximize discovering), of frontier based ones (where frontiers could be defined as occlusions). Please comment.
3. The claim of generalization is misleading. For generalization I'm expecting to train in one plant/scenario and test in a different one, maybe same species, in different scenario and environmental conditions. The need for training in the same plant that will be deployed speaks against the usefulness of the approach. Please comment.

Finally, although the problem seems interesting, I guess the real value would be in answering the question "What is behind this leaf/branch?" and not just moving everything at once.

**Quality Of The Limitations Section:**

Limitations are addressed clearly

**Questions For Rebuttal:**

See weaknesses above.

**Robotics Focus:**

Sufficient demonstration on hardware

**Summary Of Paper:**

This paper proposes a system to predict the gain, in terms of observed space, of a push action on the plant foliage. To this end, SRPNet is proposed to regress the predicted space behind the occlusions as an image, thus providing spatial information. The full system PPG uses the outcome of SRPNet to evaluate sampled actions and decide for the best action that predicts most of the space to be discovered. The networks is trained and tested in two setups, synthetic vines and a real dracaena plant.

**Summary Of Recommendation:**

Weakness overweight the interesting problem setup.

**After rebuttal**
I appreciate the effort put into the rebuttal to try answer the open points.

However,  the weaknesses I highlighted are not well addressed in my opinion. The system can be reduced to a heuristic search, with no comparisons to a information gain driven (NBV or others) approach. From field robotics deployment, the generalization capabilities are not demonstrated ( it would have been more convincing to try out a different Dracaena in an outdoor environment to show some generalization). And finally, my last point was not about moving a selected leaf, it was about the reasoning to discover, or reveal more in a sequence of decisions given the actions and observations taken so far. But, I understand the confusion from my phrasing.

All in all, I keep my rating, but as already mentioned I am fine with acceptance if the rest of the reviewers go for it.

---

> ### Author Response · Authors · 2023-08-11
> **Response to questions**
>
> Thank you for the encouraging feedback and the thoughtful suggestions. We respond to your specific questions below.
>
> 1. Yes, actions are scored using SRPNet and the most promising action is executed, and brute force evaluating all actions would scale poorly with large-action spaces. Thus, rather than brute force scoring all actions, we use the Cross-Entropy Method [54] to selectively score only a subset of the actions. More details are in Section 4.3, A.4 for vines, and B.4 for Dracaena, and we also summarize here. At a high-level, CEM is an iterative process that starts by sampling a small number of actions randomly (in our implementation 300), and then repeats the following steps for some iterations (3 in our case), before returning the best of 300 x 3 evaluated actions:
>     - evaluate sampled actions using SRPNet
>     - retain the top-k as _elite_ actions (in our case, top-60)
>     - fit Gaussian distribution to elites and sample another batch of (300) actions
>
>      This process isn’t guaranteed to return the best possible action but in-practice gives sufficiently high-performing actions. It has been widely used in past papers, e.g. [49]. Please also see related discussion on argmax action selection versus CEM in response to "Why the use of a CEM approach for action selection" for reviewer N9Cs.
>
>      Thus, use of CEM lends scalability to our proposed approach and we found it to scale well to the 2800 and 19000 actions for vines and Dracaena respectively. If action spaces grow even larger, CEM may suffer.
>
> 2. We recognize that there are works that use next best view planning for discovering hidden parts of the plan, such as [23] and other more recent works like [NBV-SC]. However, the setup in these works assume that simply moving the camera is enough to recover the hidden space. We are interested in settings where foliage needs to be moved aside to see the space behind, similar to the two examples we shown in Figure 1 left, where it is necessary to push aside the leaves to see the vegetables. For the vine setup (shown in Figure 1 (center)), due to the dense foliage, it would be difficult to just move the camera to see the surface beneath it. We view our work as tackling this alternative setting of the plant self-occlusion problem. Please also see the paragraph at L65 in the paper for a discussion of the relationship to other related work on plant manipulation (e.g. treating leaves as permeable obstacles for motion planning [29], etc).
>
> 3. Yes, we could strive for even broader definitions of generalization. However, we reiterate the generalization capability that the paper does test, which we believe is already quite non-trivial.
>     - Figure 6 shows the three settings that our vine experiments are conducted in. We train in the base setting (uniform dense vines) but also test in the sparse and separated vines setting, which we do not have any training data on. The density and configuration of the vines are different and test the robustness of SRPNet in predicting where space would be revealed.
>     - Moreover, for our real Dracaena plant, the data collection and experiment was done over multiple weeks. The plant grew in the meanwhile and the leaf dynamics changed in some ways over time. We also did not control lighting: our setup is in a room with natural sunlight, and the experiments are conducted at different times of the day. We also randomly rotated the Dracaena plant. From a physical experiment point of view, this represents substantial variation.
>     - To further test generalization, we have re-run the single action selection experiment (Table 2 in the paper) with a model trained on data from two months ago. The Dracaena plant has changed quite a bit in these two months. See the differences in the plant at https://sites.google.com/view/pushingfoliage/home?authuser=2#h.3szqmxp39t7n  The SRPNet model generalizes reasonably well, and our model retains the effectiveness at selecting good actions:
>      | Method |  mean and 95% CI |
>      |--------|-------------|
>      | Random Tangential     | 6780.7 [4399.3, 8618.1]   |
>      | PPG w/ SRPNet (Our)  | 11551.4 [9464.7, 14644.9]   |
>
> 4. We have an experiment named "Targeted Revealing Performance" (at L276, before the Conclusion section) that tests the method for revealing space behind a user-selected leaf. This experiment precisely simulates the situation where we want to see behind a particular leaf/region.
>
> Hopefully we have addressed the concerns raised in this review. We will love to hear your feedback, and we are happy to offer further clarifications or respond to any other concerns. We hope our response helps improve the impression of our work.
>
> **References:**
>
> [NBV-SC] NBV-SC: Next Best View Planning based on Shape Completion for Fruit Mapping and Reconstruction. Rohit Menon, Tobias Zaenker, Nils Dengler, Maren Bennewitz.

---

### Decision · Program_Chairs · 2023-08-30

**Decision:**

Accept (Poster)

**Comment:**

This paper focuses on manipulating plants to reveal hidden areas (e.g. moving a leaf). The method predicts revealed space for a given action and uses CEM to move the plants to reveal as much space as possible. Evaluations are conducted on two real-world plants.

Strengths:
- The presentation is clear and readable
- The method is sound and makes sense for this application
- The results demonstrate that the method is effective at revealing space behind plant parts

Weaknesses:
- It is not clear that what the method does is useful for agricultural (or other applications). Simply revealing occluded space does not seem like the correct metric. Rather, revealing specific targets (e.g. fruit or side-shoots) is what such manipulation should seek to do. As the method does not have knowledge of the overall plant structure, it seems the only way to find specific targets is to reveal as much space as possible, which is very inefficient.
- The generalization claim is overstated. It is expected that such a method transfer across a species of plants, but instead the training is plant-specific, requiring that data is collected on the specific plant to be manipulated. This is clearly not going to scale to agricultural applications. The authors should evaluate how transferable the method is across different plants. The claims about the plant growing over the time the research was conducted are not quantified and difficult to evaluate for generalization.

Overall, the rebuttal did not significantly change the reviewers' opinions.